# Fairness Through Computationally-Bounded Awareness

**Michael P. Kim**[*]
Stanford University
mpk@cs.stanford.edu

**Omer Reingold**[*]
Stanford University
reingold@stanford.edu

**Guy N. Rothblum**[†]
Weizmann Institute of Science
rothblum@alum.mit.edu

## Abstract

We study the problem of fair classification within the versatile framework of Dwork *et al.* [6], which assumes the existence of a metric that measures similarity between pairs of individuals. Unlike earlier work, we do not assume that the entire metric is known to the learning algorithm; instead, the learner can query this *arbitrary* metric a bounded number of times. We propose a new notion of fairness called *metric multifairness* and show how to achieve this notion in our setting. Metric multifairness is parameterized by a similarity metric $d$ on pairs of individuals to classify and a rich collection $\mathcal{C}$ of (possibly overlapping) "comparison sets" over pairs of individuals. At a high level, metric multifairness guarantees that *similar subpopulations are treated similarly*, as long as these subpopulations are identified within the class $\mathcal{C}$.

## 1 Introduction

More and more, machine learning systems are being used to make predictions about people. Algorithmic predictions are now being used to answer questions of significant personal consequence; for instance, *Is this person likely to repay a loan?* [24] or *Is this person likely to recommit a crime?* [1]. As these classification systems have become more ubiquitous, concerns have also grown that classifiers obtained via machine learning might discriminate based on sensitive attributes like race, gender, or sexual orientation. Indeed, machine-learned classifiers run the risk of perpetuating or amplifying historical biases present in the training data. Examples of discrimination in classification have been well-illustrated [24, 1, 5, 19, 13, 3]; nevertheless, developing a systematic approach to fairness has been challenging. Often, it feels that the objectives of fair classification are at odds with obtaining high-utility predictions.

In an influential work, Dwork *et al.* [6] proposed a framework to resolve the apparent conflict between utility and fairness, which they call "fairness through awareness." This framework takes the perspective that a fair classifier should *treat similar individuals similarly*. The work formalizes this abstract goal by assuming access to a task-specific similarity metric $d$ on pairs of individuals. The proposed notion of fairness requires that if the distance between two individuals is small, then the predictions of a fair classifier cannot be very different. More formally, for some small constant $\tau \geq 0$, we say a hypothesis $f : \mathcal{X} \rightarrow [-1, 1]$ satisfies $(d, \tau)$-*metric fairness*[1] if the following (approximate) Lipschitz condition holds for all pairs of individuals from the population $\mathcal{X}$.

$$\forall x, x' \in \mathcal{X} \times \mathcal{X} : \quad |f(x) - f(x')| \leq d(x, x') + \tau \tag{1}$$

Subject to these intuitive similarity constraints, the classifier may be chosen to maximize utility. Note that, in general, the metric may be designed externally (say, by a regulatory agency) to address legal

---

[*]Supported by NSF Grant CCF-1763299.

[†]Supported by ISF grant No. 5219/17.

[1]Note the definition given in [6] is slightly different; in particular, they propose a more general Lipschitz condition, but fix $\tau = 0$.

and ethical concerns, independent from the task of learning. In particular, in certain settings, the metric designers may have access to a different set of features than the learner. For instance, perhaps the metric designers have access to sensitive attributes, but for legal, social, or pragmatic reasons, the learner does not. In addition to its conceptual simplicity, the modularity of fairness through awareness makes it a very appealing framework. Currently, there are many (sometimes contradictory) notions of what it means for a classifier to be fair [19, 5, 9, 13, 14], and there is much debate on which definitions should be applied in a given context. Discrimination comes in many forms and classification is used in a variety of settings, so naturally, it is hard to imagine any universally-applicable definition of fairness. Basing fairness on a similarity metric offers a flexible approach for formalizing a variety of guarantees and protections from discrimination.

Still, a challenging aspect of this approach is the assumption that the similarity metric is known for all pairs of individuals.[2] Deciding on an appropriate metric is itself a delicate matter and could require human input from sociologists, legal scholars, and specialists with domain expertise. For instance, in the loan repayment example, a simple, seemingly-objective metric might be a comparison of credit scores. A potential concern, however, is that these scores might themselves be biased (i.e. encode historical discriminations). In this case, a more nuanced metric requiring human input may be necessary. Further, if the metric depends on features that are latent to the learner (e.g. some missing sensitive feature) then the metric could appear *arbitrarily complex* to the learner. As such, in many realistic settings, the resulting metric will not be a simple function of the learner's feature vectors of individuals.

In most machine learning applications, where the universe of individuals is assumed to be very large, even writing down an appropriate metric could be completely infeasible. In these cases, rather than require the metric value to be specified for all pairs of individuals, we could instead ask a panel of experts to provide similarity scores for a *small sample* of pairs of individuals. While it is information-theoretically impossible to guarantee metric fairness from a sampling-based approach, we still might hope to provide a strong, provable notion of fairness that maintains the theoretical appeal and practical modularity of the fairness through awareness framework.

In this work, we propose a new theoretical framework for fair classification based on fairness through awareness – which we dub "fairness through computationally-bounded awareness" – that eliminates the considerable issue of requiring the metric to be known exactly. Our approach maintains the simplicity and flexibility of fairness through awareness, but provably only requires a small number of random samples from the underlying metric, even though we make no structural assumptions about the metric. In particular, our approach works even if the metric provably cannot be learned. Specifically, our notion will require that a fair classifier *treat similar subpopulations of individuals similarly*, in a sense we will make formal next. While our definition relaxes fairness through awareness, we argue that it still protects against important forms of discrimination that the original work aimed to combat; further, we show that stronger notions necessarily require a larger sample complexity from the metric. As in [6], we investigate how to learn a classifier that achieves optimal utility under similarity-based fairness constraints, assuming a weaker model of limited access to the metric. We give positive and negative results that show connections between achieving our fairness notion and learning.

## 2 Setting and metric multifairness

**Notation.** Let $\mathcal{X} \subseteq \mathbb{R}^n$ denote the universe over individuals we wish to classify, where $x \in \mathcal{X}$ encodes the features of an individual. Let $\mathcal{D}$ denote the data distribution over individuals and labels supported on $\mathcal{X} \times \{-1, 1\}$; denote by $x, y \sim \mathcal{D}$ a random draw from this distribution. Additionally, let $\mathcal{M}$ denote the metric sample distribution over pairs of individuals. For a subset $S \subseteq \mathcal{X} \times \mathcal{X}$, we denote by $(x, x') \sim S$ a random draw from the distribution $\mathcal{M}$ conditioned on $(x, x') \in S$. Let $d : \mathcal{X} \times \mathcal{X} \to [0, 2]$ denote the underlying fairness metric that maps pairs to their associated distance.[3]

Our learning objective will be to minimize the expectation over $\mathcal{D}$ of some convex loss function $L : [-1, 1] \times [-1, 1] \to \mathbb{R}_+$ over a convex hypothesis class $\mathcal{F}$, subject to the fairness constraints. We focus on agnostically learning the hypothesis class of linear functions with bounded weight; for some constant $B > 0$, let $\mathcal{F} = [-B, B]^n$. For $w \in \mathcal{F}$, define $f_w(x) = \langle w, x \rangle$, projecting $f_w(x)$

onto $[-1, 1]$ to get a valid prediction. We assume $\|x\|_1 \leq 1$ for all $x \in \mathcal{X}$; this is without loss of generality, by normalizing and increasing $B$ appropriately.

The focus on linear functions is not too restrictive; in particular, by increasing the dimension to $n' = O(n^k)$, we can learn any degree-$k$ polynomial function of the original features. By increasing $k$, we can approximate increasingly complex functions.

## 2.1 Metric multifairness

We define our relaxation of metric fairness with respect to a rich class of statistical tests on the pairs of individuals. Let a *comparison* be any subset of the pairs of $\mathcal{X} \times \mathcal{X}$. Our definition, which we call *metric multifairness*, is parameterized by a collection of comparisons $\mathcal{C} \subseteq 2^{\mathcal{X} \times \mathcal{X}}$ and requires that a hypothesis appear Lipschitz according to all of the statistical tests defined by the comparisons $S \in \mathcal{C}$.

**Definition** (Metric multifairness). *Let $\mathcal{C} \subseteq 2^{\mathcal{X} \times \mathcal{X}}$ be a collection of comparisons and let $d : \mathcal{X} \times \mathcal{X} \to [0, 2]$ be a metric. For some constants $\tau \geq 0$, a hypothesis $f$ is $(\mathcal{C}, d, \tau)$-metric multifair if for all $S \in \mathcal{C}$,*

$$\mathop{\mathbf{E}}_{(x,x') \sim S} \big[ |f(x) - f(x')| \big] \leq \mathop{\mathbf{E}}_{(x,x') \sim S} \big[ d(x, x') \big] + \tau. \tag{2}$$

To begin, note that metric multifairness is indeed a relaxation of metric fairness; if we take the collection $\mathcal{C} = \{\{(x, x')\} : x, x' \in \mathcal{X} \times \mathcal{X}\}$ to be the collection of all pairwise comparisons, then $(\mathcal{C}, d, \tau)$-metric multifairness is equivalent to $(d, \tau)$-metric fairness.

In order to achieve metric multifairness from a small sample from the metric, however, we need a lower bound on the density of each comparison in $\mathcal{C}$; in particular, we can't hope to enforce metric fairness from a small sample. For some $\gamma > 0$, we say that a collection of comparisons $\mathcal{C}$ is $\gamma$-*large* if for all $S \in \mathcal{C}$, $\Pr_{(x,x') \sim \mathcal{M}}[(x, x') \in S] \geq \gamma$. A natural next choice for $\mathcal{C}$ would be a collection of comparisons that represent the Cartesian products between traditionally-protected groups, defined by race, gender, etc. In this case, as long as the minority populations are not too small, then a random sample from the metric will concentrate around the true expectation, and we could hope to enforce this statistical relaxation of metric fairness. While this approach is information-theoretically feasible, its protections are very weak.

To highlight this weakness, suppose we want to predict the probability individuals will repay a loan, and our metric is an adjusted credit score. Even after adjusting scores, two populations $P, Q \subseteq \mathcal{X}$ (say, defined by race) may have large average distance because *overall* $P$ has better credit than $Q$; still, within $P$ and $Q$, there may be significant *subpopulations* $P' \subseteq P$ and $Q' \subseteq Q$ that should be treated similarly (possibly representing the qualified members of each group). In this case, a coarse statistical relaxation of metric fairness will not require that a classifier treat $P'$ and $Q'$ similarly; instead, the classifier could treat everyone in $P$ better than everyone in $Q$ – including treating *unqualified* members of $P$ better than *qualified* members of $Q$. Indeed, the weaknesses of broad-strokes statistical definitions served as motivation for the original work of [6]. We would like to choose a class $\mathcal{C}$ that strengthens the fairness guarantees of metric multifairness, but maintains its efficient sample complexity.

**Computationally-bounded awareness.** While we can define metric multifairness with respect to any collection $\mathcal{C}$, typically, we will think of $\mathcal{C}$ as a rich class of overlapping subsets; equivalently, we can think of the collection $\mathcal{C}$ as an expressive class of boolean functions, where for $S \in \mathcal{C}$, $c_S(x, x') = 1$ if and only if $(x, x') \in S$. In particular, $\mathcal{C}$ should be much more expressive than simply defining comparisons across traditionally-protected groups. The motivation for choosing such an expressive class $\mathcal{C}$ is exemplified in the following proposition.

**Proposition 1.** *Suppose there is some $S \in \mathcal{C}$, such that $\mathbf{E}_{(x,x') \sim S}[d(x, x')] \leq \varepsilon$. Then if $f$ is $(\mathcal{C}, d, \tau)$-metric multifair, then $f$ satisfies $(d, (\varepsilon + \tau)/p)$-metric fairness for at least a $(1 - p)$-fraction of the pairs in $S$.*

That is, if there is some subset $S \in \mathcal{C}$ that identifies a set of pairs whose metric distance is small, then any metric multifair hypothesis must also satisfy the stronger individual metric fairness notion on many pairs from $S$. This effect will compound if many different (possibly overlapping) comparisons are identified that have small average distance. We emphasize that these small-distance comparisons are not known before sampling from the metric; indeed, this would imply the metric was (approximately) known *a priori*. Still, if the class $\mathcal{C}$ is rich enough to correlate well with various comparisons that

reveal significant information about the metric, then any metric multifair hypothesis will satisfy *individual-level* fairness on a significant fraction of the population!

While increasing the expressiveness of $\mathcal{C}$ increases the strength of the fairness guarantee, in order to learn from a small sample, we cannot choose $\mathcal{C}$ to be arbitrarily complex. Thus, in choosing $\mathcal{C}$ we must balance the strength of the fairness guarantee with the information bottleneck in accessing $d$ through random samples. Our resolution to these competing needs is complexity-theoretic: while information-theoretically, we can't hope to ensure fair treatment across *all* subpopulations, we can hope ensure fair treatment across *efficiently-identifiable* subpopulations. For instance, if we take $\mathcal{C}$ to be a family defined according to some class of computations of bounded dimension – think, the set of conjunctions of a constant number of boolean features or short decision trees – then we can hope to accurately estimate and enforce the metric multifairness conditions. Taking such a bounded $\mathcal{C}$ ensures that a hypothesis will be fair on all comparisons identifiable within this computational bound. This is the sense in which metric multifairness provides fairness through *computationally-bounded* awareness.

## 2.2 Learning model

**Metric access.**  Throughout, our goal is to learn a hypothesis from noisy samples from the metric that satisfies multifairness. Specifically, we assume an algorithm can obtain a small number of independent random *metric samples* $(x, x', \Delta(x, x')) \in \mathcal{X} \times \mathcal{X} \times [0, 2]$ where $(x, x') \sim \mathcal{M}$ is drawn at random over the distribution of pairs of individuals, and $\Delta(x, x')$ is a random variable of bounded magnitude with $\mathbf{E}[\Delta(x, x')] = d(x, x')$.

We emphasize that this is a very limited access model. As Theorem 2 shows we achieve $(\mathcal{C}, d, \tau)$-metric multifairness from a number of samples that depends *logarithmically* on the size of $\mathcal{C}$ independent of the complexity of the similarity metric.[4] Recall that $d : \mathcal{X} \times \mathcal{X} \to [0, 2]$ can be an *arbitrary* symmetric function; thus, the learner does not necessarily have enough information to learn $d$. Still, for exponentially-sized $\mathcal{C}$, the learner can guarantee metric multifairness from a polynomial-sized sample, and the strength of the guarantee will scale up with the complexity of $\mathcal{C}$ (as per Propsition 1).

In order to ensure a strong notion of fairness, we assume that the subpopulations we wish to protect are well-represented in the pairs drawn from $\mathcal{M}$. This assumption, while important, is not especially restrictive, as we think of the metric samples as coming from a regulatory committee or ethically-motivated party; in other words, in practical settings, it is reasonable to assume that one can choose the metric sampling distribution based on the notion of fairness one wishes to enforce.

**Label access.**  When we learn linear families, our goal will be to learn from a sample of labeled examples. We assume the algorithm can ask for independent random samples $x, y \sim \mathcal{D}$.

**Measuring optimality.**  To evaluate the utility guarantees of our learned predictions, we take a comparative approach. Suppose $\mathcal{H} \subseteq 2^{\mathcal{X} \times \mathcal{X}}$ is a collection of comparisons. For $\varepsilon \geq 0$, we say a hypothesis $f$ is $(\mathcal{H}, \varepsilon)$-*optimal* with respect to $\mathcal{F}$, if

$$\mathop{\mathbf{E}}_{x,y \sim \mathcal{D}}[L(f(x), y)] \leq \mathop{\mathbf{E}}_{x,y \sim \mathcal{D}}[L(f^*(x), y)] + \varepsilon \tag{3}$$

where $f^* \in \mathcal{F}$ is an optimal $(\mathcal{H}, d, 0)$-metric multifair hypothesis.

## 3   Learning a metric multifair hypothesis

As in [6], we formulate the problem of learning a fair set of predictions as a convex program. Our objective is to minimize the expected loss $\mathbf{E}_{x,y \sim \mathcal{D}}[L(f(x), y)]$, subject to the multifairness constraints defined by $\mathcal{C}$.[5] Specifically, we show that a simple variant of stochastic gradient descent due to [20] learns such linear families efficiently.

**Theorem 2.** *Suppose $\gamma, \tau, \delta > 0$ and $\mathcal{C} \subseteq 2^{\mathcal{X} \times \mathcal{X}}$ is $\gamma$-large. With probability at least $1 - \delta$, stochastic switching subgradient descent learns a hypothesis $w \in \mathcal{F}$ that is $(\mathcal{C}, d, \tau)$-metric multifair and $(\mathcal{C}, O(\tau))$-optimal with respect to $\mathcal{F}$ in $O\left(\frac{B^2 n^2 \log(n/\delta)}{\tau^2}\right)$ iterations from $m = \tilde{O}\left(\frac{\log(|\mathcal{C}|/\delta)}{\gamma \tau^2}\right)$ metric samples. Each iteration uses at most $1$ labeled example and can be implemented in $\tilde{O}\left(|C| \cdot n \cdot \operatorname{poly}(1/\gamma, 1/\tau, \log(1/\delta))\right)$ time.*

Note that the metric sample complexity depends logarithmically on $|\mathcal{C}|$. Thus, information-theoretically, we can hope to enforce metric mutlifairness with a class $\mathcal{C}$ that grows exponentially and still be efficient. While the running time of each iteration depends on $|\mathcal{C}|$, note that the number of iterations is *independent* of $|\mathcal{C}|$. In Section 4, we show conditions on $|\mathcal{C}|$ under which we can speed up the running time of each iteration to depend logarithmically on $|\mathcal{C}|$.

We give a description of the switching subgradient method in Algorithm 1. At a high level, at each iteration, the procedure checks to see if any constraint is significantly violated. If it finds a violation, it takes a (stochastic) step towards feasibility. Otherwise, it steps according a stochastic subgradient for the objective.

For convenience of analysis, we define the residual on the constraint defined by $S$ as follows.

$$R_S(w) = \mathop{\mathbf{E}}_{(x,x') \sim S}\left[\left||f_w(x) - f_w(x')|\right|\right] - \mathop{\mathbf{E}}_{(x,x') \sim S}\left[d(x,x')\right] \tag{4}$$

Note that $R_S(w)$ is convex in the predctions $f_w(x)$ and thus, for linear families is convex in $w$. We describe the algorithm assuming access to the following estimators, which we can implement efficiently (in terms of time and samples). First, we assume we can estimate the residual $\hat{R}_S(w)$ on each $S \in \mathcal{C}$ with tolerance $\tau$ such that for all $w \in \mathcal{F}$, $\left|R_S(w) - \hat{R}_S(w)\right| \leq \tau$. Next, we assume access to a stochastic subgradient oracle for the constraints and the objective. For a function $\phi(w)$, let $\partial \phi(w)$ denote the set of subgradients of $\phi$ at $w$. We abuse notation, and let $\nabla \phi(w)$ refer to a vector-valued random variable where $\mathbf{E}[\nabla \phi(w)|w] \in \partial \phi(w)$. We assume access to stochastic subgradients for $\partial R_S(w)$ for all $S \in \mathcal{C}$ and $\partial L(w)$. We include a full analysis of the algorithm and proof of Theorem 2 in the Appendix.

### 3.1 Post-processing for metric multifairness

One specific application of this result is as a way post-process learned predictions to ensure fairness. In particular, suppose we are given the predictions from some pre-trained model for $N$ individuals, but are concerned that these predictions may not be fair. We can use Algorithm 1 to post-process these labels to select near-optimal metric multifair predictions. Note these predictions will be optimal with respect to the *unconstrained* family of predictions – not just predictions that come from a specific hypothesis class (like linear functions).

Specifically, in this setting we can represent an unconstrained set of predictions as a linear hypothesis in $N$ dimensions: take $B = 1$, and let the feature vector for $x_i \in \mathcal{X}$ be the $i$th standard basis vector.

**Algorithm 1:** Switching Subgradient Descent

---
Let $\tau > 0$, $T \in \mathbb{N}$, and $\mathcal{C} \subseteq 2^{\mathcal{X} \times \mathcal{X}}$.
Initialize $w_0 \in \mathcal{F} = [-B, B]^n$; $W = \emptyset$
For $k = 1, \ldots, T$:

- If $\exists S \in \mathcal{C}$ such that $\hat{R}_S(w_k) > 4\tau/5$:        `// some constraint violated`

    - $S_k \leftarrow$ any $S \in \mathcal{C}$ such that $\hat{R}_S(w_k) > 4\tau/5$
    - $w_{k+1} \leftarrow w_k - \frac{\tau}{M^2} \nabla R_{S_k}(w_k)$     `/* step according to constraint`
      `project onto F if necessary */`

- Else:                                    `// no violations found`

    - $W \leftarrow W \cup \{w_k\}$          `// update set of feasible iterates`
    - $w_{k+1} \leftarrow w_k - \frac{\tau}{GM} \nabla L(w_k)$      `/* step according to objective`
      `project onto F if necessary */`

Output $\bar{w} = \frac{1}{|W|} \cdot \sum_{w \in W} w$        `// output average of feasible iterates`

---

Then, we can think of the input labels to Algorithm 1 to be the *output* of any predictor that was trained separately.[6] For instance, if we have learned a highly-accurate model, but are unsure of its fairness, we can instantiate our framework with, say, the squared loss between the original predictions and the returned predictions; then, we can view the program as a procedure to *project* the highly-accurate predictions onto the set of metric multifair predictions. Importantly, our procedure only needs a small set of samples from the metric and *not* the original data used to train the model.

Post-processing prediction models for fairness has been studied in a few contexts [25, 14, 18]. This post-processing setting should be contrasted to these settings. In our setting, the predictions are not required to generalize out of sample (in terms of loss or fairness). On the one hand, this means the metric multifairness guarantee does not generalize outside the $N$ individuals; on the other hand, because the predictions need not come from a bounded hypothesis class, their utility can only improve compared to learning a metric multifair hypothesis directly.

In addition to preserving the utility of previously-trained classifiers, separating the tasks of training for utility and enforcing fairness may be desirable when intentional malicious discrimination may be anticipated. For instance, when addressing the forms of racial profiling that can occur through targeted advertising (as described in [6]), we may not expect self-interested advertisers to adhere to classification under strict fairness constraints, but it stands to reason that prominent advertising platforms might want to prevent such blatant abuses of their platform. In this setting, the platform could impose metric multifairness after the advertisers specify their ideal policy.

## 4    Reducing search to agnostic learning

As presented above, the switching subgradient descent method converges to a nearly-optimal point in a bounded number of subgradient steps, independent of $|\mathcal{C}|$. The catch is that at the beginning of each iteration, we need to search over $\mathcal{C}$ to determine if there is a significantly violated multifairness constraint. As we generally want to take $\mathcal{C}$ to be a rich class of comparisons, in many cases $|\mathcal{C}|$ will be prohibitive. As such, we would hope to find violated constraints in sublinear time, preferably even poly-logarithmic in $|C|$. Indeed, we show that if a concept class $\mathcal{C}$ admits an efficient agnostic learner, then we can solve the violated constraint search problem over the corresponding collection of comparisons efficiently.

Agnostic learning can be phrased as a problem of detecting correlations. Suppose $g, h : \mathcal{U} \to [-1, 1]$, and let $\mathcal{D}$ be some distribution supported on $\mathcal{U}$. We denote the correlation between $g$ and $h$ on $\mathcal{D}$ as $\langle g, h \rangle = \mathbf{E}_{i \sim \mathcal{D}}[g(i) \cdot h(i)]$. We let $\mathcal{C} \subseteq [-1, 1]^{\mathcal{U}}$ denote the *concept class* and $\mathcal{H} \subseteq [-1, 1]^{\mathcal{U}}$ denote the *hypothesis class*. The task of agnostic learning can be stated as follows: given sample access over some distribution $(i, g(i)) \sim \mathcal{D} \times [-1, 1]$ to some function $g \in [-1, 1]^N$, find some hypothesis $h \in \mathcal{H}$ that is comparably correlated with $g$ as the best $c \in \mathcal{C}$. That is, given access to $g$, an agnostic learner with accuracy $\varepsilon$ for concept class $\mathcal{C}$ returns some $h$ from the hypothesis class $\mathcal{H}$ such that

$$\langle g, h \rangle + \varepsilon \geq \max_{c \in \mathcal{C}} \langle g, c \rangle. \tag{5}$$

An agnostic learner is typically considered efficient if it runs in polynomial time in $\log(|\mathcal{C}|)$ (or an appropriate notion of dimension of $\mathcal{C}$), $1/\varepsilon$, and $\log(1/d)$. Additionally, distribution-specific learners and learners with query access to the function have been studied [12, 7]. In particular, membership queries tend to make agnostic learning easier. Our reduction does not use any metric samples other than those that the agnostic learner requests. Thus, if we are able to query a panel of experts according to the learner, rather than randomly, then an agnostic learner that uses queries could be used to speed up our learning procedure.

**Theorem 3.** *Suppose there is an algorithm $\mathcal{A}$ for agnostic learning the concept class $\mathcal{C}$ with hypothesis class $\mathcal{H}$ that achieves accuracy $\varepsilon$ with probability $1 - \delta$ in time $T_{\mathcal{A}}(\varepsilon, \delta)$ from $m_{\mathcal{A}}(\varepsilon, \delta)$ labeled samples. Suppose that $\mathcal{C}$ is $\gamma$-large. Then, there is an algorithm that, given access to $T = \tilde{O}\left(\frac{B^2 n^2}{\tau^2}\right)$ labeled examples, outputs a set of predictions that are $(\mathcal{C}, d, \tau)$-metric multifair and $(\mathcal{H}, O(\tau))$-optimal with respect to $\mathcal{F} = [-B, B]^n$ that runs in time $\tilde{O}\left(\frac{T_{\mathcal{A}}(\gamma\tau, \delta/T) \cdot B^2 n^2}{\gamma^2 \tau^2}\right)$ and requires $m = \tilde{O}\left(\frac{\log(|\mathcal{C}|)}{\gamma\tau^2} + \frac{n_{\mathcal{A}}(\gamma\tau, \delta/T)}{\gamma\tau^2}\right)$ metric samples.*

When we solve the convex program with switching subgradient descent, at the beginning of each iteration, we check if there is any $S \in \mathcal{C}$ such that the residual quantity $R_S(w)$ is greater than some threshold. If we find such an $S$, we step according to the subgradient of $R_S(w)$. In fact, the proof of the convergence of switching subgradient descent reveals that as long as when there is some $S \in \mathcal{C}$ where $R_S(w)$ is in violation, we can find some $R_{S'}(w) > \rho$ for some constant $\rho$, where $S' \in \mathcal{H} \subseteq [-1, 1]^{\mathcal{X} \times \mathcal{X}}$, then we can argue that the learned hypothesis will be $(\mathcal{C}, d, \tau)$-metric multifair and achieve utility commensurate with the best $(\mathcal{H}, d, 0)$-metric multifair hypothesis.

We show a general reduction from the problem of searching for a violated comparison $S \in \mathcal{C}$ to the problem of agnostic learning over the corresponding family of boolean functions. In particular, recall that for a collection of comparisons $\mathcal{C} \subseteq 2^{\mathcal{X} \times \mathcal{X}}$, we can also think of $\mathcal{C}$ as a family of boolean concepts, where for each $S \in \mathcal{C}$, there is an associated boolean function $c_S : \mathcal{X} \times \mathcal{X} \to \{-1, 1\}$ where $c_S(x_i, x_j) = 1$ if and only if $(x_i, x_j) \in S$. We frame this search problem as an agnostic learning problem, where we design a set of "labels" for each pair $(x, x') \in \mathcal{X} \times \mathcal{X}$ such that any function that is highly correlated with these labels encodes a way to update the parameters towards feasibility.

*Proof.* Recall the search problem: given a current hypothesis $f_w$, is there some $S \in \mathcal{C}$ such that $R_S(w) = \mathbf{E}_{(x,x') \sim S}[|f_w(x_i) - f_w(x_j)| - d(x_i, x_j)] > \tau$? Consider the labeling each pair $(x_i, x_j)$ with $v(x_i, x_j) = |f_w(x_i) - f_w(x_j)| - d(x_i, x_j)$. Let $\rho = R_{\mathcal{X} \times \mathcal{X}}(w)$; note that we can treat $\rho$ as a constant for all $S \in \mathcal{C}$. Further, suppose $S \in \mathcal{C}$ is such that $R_S(w) > \tau$, or equivalently, $\mathbf{E}_{(x_i, x_j) \sim S}[v(x_i, x_j)] > \tau$. Then, by the assumption that $\mathcal{C}$ is $\gamma$-large, the correlation between the corresponding boolean function $c_S$ and labels $v$ can be lower bounded as $\langle c_S, v \rangle > 2\gamma\tau - \rho$. Suppose we have an agnostic learner that returns a hypothesis $h$ with $\varepsilon < \gamma\tau$ accuracy. Then, we know that $\langle h, v \rangle \geq \gamma\tau - \rho$ by the lower bound on the optimal $c_S \in \mathcal{C}$. Then, consider the function $R_h(w)$ defined as follows.

$$R_{S_h}(w) = \mathop{\mathbf{E}}_{(x,x') \sim \mathcal{X} \times \mathcal{X}} \left[ \left( \frac{h(x, x') + 1}{2} \right) \cdot v(x, x') \right] \tag{6}$$

$$= \frac{\langle h, v \rangle + \rho}{2} \tag{7}$$

Thus, given that there exists some $S \in \mathcal{C}$ where $R_S(w) > \tau$, we can find some real-valued comparison $S_h(x, x') = \frac{h(x,x')+1}{2}$, such that $R_{S_h}(w) = \Omega(\langle h, v \rangle + \rho) \geq \Omega(\gamma\tau)$. $\qquad\square$

Discovering a violation of at least $\Omega(\gamma\tau)$ guarantees $\Omega(\gamma^2\tau^2)$ progress in the duality gap at each step, so the theorem follows from the analysis of Algorithm 1.

## 5  Hardness of learning metric multifair hypotheses

In this section, we show that our algorithmic results cannot be improved significantly. In particular, we focus on the post-processing setting of Section 3.1. We show that the metric sample complexity is tight up to a $\Omega(\log \log(|\mathcal{C}|))$ factor unconditionally. We also show that some learnability assumption on $\mathcal{C}$ is necessary in order to achieve a high-utility $(\mathcal{C}, d, \tau)$-metric multifair predictions efficiently. In particular, we give a reduction from inverting a boolean concept $c \in \mathcal{C}$ to learning a hypothesis $f$ that is metric multifair on a collection $\mathcal{H}$ derived from $\mathcal{C}$, where the metric samples from $d$ encode information about the concept $c$. Recall, that for any $\mathcal{H}$ and $d$, we can always output a trivial $(\mathcal{H}, d, 0)$-metric multifair hypothesis by outputting a constant hypothesis. This leads to a subtlety in our reductions, where we need to leverage the learner's ability to simultaneously satisfy the metric multifairness constraints and achieve high utility.

Both lower bounds follow the same general construction. Suppose we have some boolean concept class $\mathcal{C} \subseteq \{-1, 1\}^{\mathcal{X}_0}$ for some universe $\mathcal{X}_0$. We will construct a new universe $\mathcal{X} = \mathcal{X}_0 \cup \mathcal{X}_1$ and define a collection of "bipartite" comparisons over subsets of $\mathcal{X}_0 \times \mathcal{X}_1$. Then, given samples from $(x_0, c(x_0))$, we define corresponding metric values where $d(x_0, x_1)$ is some function of $c(x_0)$ for all $x_1 \in \mathcal{X}_1$. Finally, we need to additionally encode the objective of inverting $c$ into labels for $x_0 \in \mathcal{X}_0$, such that to obtain good loss, the post-processor must invert $c$ on $\mathcal{X}_0$. We give a full description of the reduction in the Appendix.

**Lower bounding the sample complexity.** While we argued earlier that *some* relaxation of metric fairness is necessary if we want to learn from a small set of metric samples, it is not clear that multifairness with respect to $\mathcal{C}$ is the strongest relaxation we can obtain. In particular, we might hope to guarantee fairness on *all* large comparisons, rather than just a finite class $\mathcal{C}$. The following theorem shows that such a hope is misplaced: in order for an algorithm to guarantee that the Lipschitz condition holds in expectation over a finite collection of large comparisons $\mathcal{C}$, then either the algorithm takes $\Omega(\log|\mathcal{C}|)$ random metric samples, or the algorithm outputs a set of nearly useless predictions. For concreteness, we state the theorem in the post-processing setting of Section 3.1; the construction can be made to work in the learning setting as well.

**Theorem 4.** *Let $\gamma, \tau > 0$ be constants and suppose $\mathcal{A}$ is an algorithm that has random sample access to $d$ and outputs a $(\mathcal{C}, d, \tau)$-metric multifair set of predictions for $\gamma$-large $\mathcal{C}$. Then, $\mathcal{A}$ takes $\Omega(\log|\mathcal{C}|)$ random samples from $d$ or outputs a set of predictions with loss that approaches the loss achievable with no metric queries.*

The construction uses a reduction from the problem of learning a linear function; we then appeal to a lower bound from linear algebra on the number of random queries needed to span a basis.

**Hardness from pseudorandom functions.** Our reduction implies that a post-processing algorithm for $(\mathcal{C}, d, \tau)$-metric multifairness with respect to an arbitrary metric $d$ gives us a way of distinguishing functions in $\mathcal{C}$ from random.

**Proposition 5** (Informal)**.** *Assuming one-way functions exist, there is no efficient algorithm for computing $(\mathcal{C}, \tau)$-optimal $(\mathcal{C}, d, \tau)$-metric multifair predictions for general $\mathcal{C}, d$, and constant $\tau$.*

Essentially, without assumptions that $\mathcal{C}$ is a learnable class of boolean functions, some nontrivial running time dependence on $|\mathcal{C}|$ is necessary. The connection between learning and pseudorandom functions [23, 11] is well-established; under stronger cryptographic assumptions as in [2], the reduction implies that a running time of $\Omega(|\mathcal{C}|^{\alpha})$ is necessary for some constant $\alpha > 0$.

## 6   Related works and discussion

Many exciting recent works have investigated fairness in machine learning. In particular, there is much debate on the very definitions of what it means for a classifier to be fair [19, 4, 21, 13, 5, 14]. Beyond the work of Dwork *et al.* [6], our work bears most similarity to two recent works of Hébert-Johnson *et al.* and Kearns *et al.* [14, 16]. As in this work, both of these papers investigate notions of fairness that aim to strengthen the guarantees of statistical notions, while maintaining their practicality. These works also both draw connections between achieving notions of fairness and efficient agnostic learning. In general, agnostic learning is considered a notoriously hard computational problem [15, 17, 8]; that said, in the context of fairness in machine learning, [16] show that using heuristic methods to agnostically learn linear hypotheses seems to work well in practice.

Metric multifairness does not directly generalize either [14] or [16], but we argue that it provides a more flexible alternative to these approaches for subpopulation fairness. In particular, these works aim to achieve specific notions of fairness – either calibration or equalized error rates – across a rich class of subpopulations. As has been well-documented [19, 4, 21], calibration and equalized error rates, in general, cannot be simultaneously satisfied. Often, researchers frame this incompatibility as a choice: either you satisfy calibration or you satisfy equalized error rates; nevertheless, there are many applications where some interpolation between accuracy (à la calibration) and corrective treatment (à la equalized error rates) seems appropriate.

Metric-based fairness offers a way to balance these conflicting fairness desiderata. In particular, one could design a similarity metric that preserves accuracy in predictions and separately a metric that performs corrective treatment, and then enforce metric multifairness on an appropriate combination of the metrics. For instance, returning to the loan repayment example, an ideal metric might be a combination of credit scores (which tend to be calibrated) and a metric that aims to increase the loans given to historically underrepresented populations (by, say, requiring the top percentiles of each subpopulation be treated similarly). Different combinations of the two metrics would place different weights on the degree of calibration and corrective discrimination in the resulting predictor. Of course, one could equally apply this metric in the framework of [6], but the big advantage with metric multifairness is that we only need a small sample from the metric to provide a relaxed, but still strong guarantee of fairness.

We are optimistic that metric multifairness will provide an avenue towards implementing metric-based fairness notions. At present, the results are theoretical, but we hope this work can open the door to empirical studies across diverse domains, especially since one of the strengths of the framework is its generality. We view testing the empirical performance of metric multifairness with various choices of metric $d$ and collection $\mathcal{C}$ as an exciting direction for future research.

Finally, two recent theoretical works also investigate extensions to the fairness through awareness framework of [6]. Gillen *et al.* [10] study metric-based individually fair online decision-making in the presence of an unknown fairness metric. In their setting, every day, a decision maker must choose between candidates available on that day; the goal is to have the decision maker's choices appear metric fair on each day (but not across days). Their work makes a strong learnability assumption about the underlying metric; in particular, they assume that the unknown metric is a Mahalanobis metric, whereas our focus is on fair classification when the metric is unknown and unrestricted. Rothblum and Yona [22] study fair machine learning under a different relaxation of metric fairness, which they call *approximate* metric fairness. They assume that the metric is fully specified and known to the learning algorithm, whereas our focus is on addressing the challenge of an unknown metric. Their notion of approximate metric fairness aims to protect *all* (large enough) groups, and thus, is more strict than metric multifairness.

**Acknowledgements.** *The authors thank Cynthia Dwork, Roy Frostig, Fereshte Khani, Vatsal Sharan, Paris Siminelakis, and Gregory Valiant for helpful conversations and feedback on earlier drafts of this work. We thank the anonymous reviewers for their careful reading and suggestions on how to improve the clarity of the presentation.*

## Footnotes

[2]Indeed, [6] identifies this assumption as "one of the most challenging aspects" of the framework.

[3]In fact, all of our results hold for a more general class of non-negative symmetric distance functions.

[4]Alternatively, for continuous classes of $\mathcal{C}$, we can replace $\log(|\mathcal{C}|)$ with some notion of dimension (VC-dimension, metric entropy, etc.) through a uniform convergence argument.

[5]For the sake of presentation, throughout the theorem statements, we will assume that $L$ is $O(1)$-Lipschitz on the domain of legal predictions/labels to guarantee bounded error; our results are proved more generally.

[6]Nothing in our analysis required labels $y \in \{-1, 1\}$; we can instead take the labels $y \in [-1, 1]$.

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
