[Supplementary Material · ftba-NeurIPS-app.pdf]

# A  Analysis of Algorithm 1

**Overview of analysis**  Here, we give a high-level overview of the analysis of Algorithm 1. We defer some technical lemmas to Appendix A.3. We refer to $K_f$ as the set of "feasible iterations" where we step according to the objective; that is,

$$K_f = \left\{ k \in [T] : \hat{R}_{S_k}(w_k) \leq 4\tau/5 \right\} \tag{8}$$

**Fairness analysis**  We begin by showing that the hypothesis $\bar{w}$ that Algorithm 1 returns satisfies metric multifairness.

**Lemma 6.** *Suppose for all $S \in \mathcal{C}$, the residual oracle $\hat{R}_S$ has tolerance $\tau/5$. Then, $\bar{w}$ is $(\mathcal{C}, d, \tau)$-metric multifair.*

*Proof.* We choose our final hypothesis $\bar{w}$ to be the weighted average of the feasible iterates. Note that the update rules for $K_f$ and $W$ imply that $\bar{w}$ is a convex combination of hypotheses where no constraint appears significantly violated, $\bar{w} = \frac{1}{|K_f|} \cdot \sum_{k \in K_f} w_k$. By convexity of $R_S$ we have the following inequality for all $S \in \mathcal{C}$.

$$R_S(\bar{w}) = R_S \left( \frac{1}{|K_f|} \sum_{k \in K_f} w_k \right) \leq \frac{1}{|K_f|} \sum_{k \in K_f} R_S(w_k) \tag{9}$$

Further, for all $S \in \mathcal{C}$ and all $k \in [T]$, by the assumed tolerance of $R_S$, we know that

$$\left| R_S(w_k) - \hat{R}_S(w_k) \right| \leq \tau/5.$$

Given that for all $k \in K_f$, $\hat{R}_{S_k}(w_k) \leq 4\tau/5$, then applying the triangle inequality, we conclude that for each comparison $S \in \mathcal{C}$,

$$\mathop{\mathbf{E}}_{(x,x') \sim S} \left[ |f_{\bar{w}}(x) - f_{\bar{w}}(x')| - d(x, x') \right] = R_S(\bar{w}) \leq \tau.$$

Hence, $\bar{w}$ is $(\mathcal{C}, d, \tau)$-metric multifair.  $\square$

**Utility and runtime analysis**  We analyze the utility of Algorithm 1 using a duality argument. For notational convenience, denote $L(w) = \mathbf{E}_{x_i \sim \mathcal{D}}[L(f_w(x_i), y_i)]$. In addition to the assumptions in the main body, throughout, we assume the following bounds on the subgradients for all $w \in \mathcal{F}$.

$$\forall S \in \mathcal{C} : \|\nabla R_S(w)\|_\infty \leq m \qquad\qquad \|\nabla L(w)\|_\infty \leq g \tag{10}$$

Assuming an $\ell_\infty$ bound implies a bound on the corresponding second moments of the stochastic subgradients; specifically, we use the notation $\|\nabla R_S(w)\|_2^2 \leq M^2 = m^2 n$ and $\|\nabla L(w)\|_2^2 \leq G^2 = g^2 n$.

Consider the Lagrangian of the program $\mathcal{L} : \mathcal{F} \times \mathbb{R}_+^{|\mathcal{C}|} \to \mathbb{R}$.

$$\mathcal{L}(w, \lambda) = L(w) + \sum_{S \in \mathcal{C}} \lambda_S R_S(w) \tag{11}$$

Let $w_* \in \mathcal{F}$ be an optimal feasible hypothesis; that is, $w_*$ is a $(\mathcal{C}, d, 0)$-metric multifair hypothesis such that $L(w_*) \leq L(w)$ for all other $(\mathcal{C}, d, 0)$-metric multifair hypotheses $w \in \mathcal{F}$.[7] By its optimality and feasibility, we know that $w_*$ achieves objective value $L(w_*) = \inf_{w \in F} \sup_{\lambda \in \mathbb{R}_+^{|\mathcal{C}|}} \mathcal{L}(w, \lambda)$.

Recall, the dual objective is given as $D(\lambda) = \inf_{w \in \mathcal{F}} \mathcal{L}(w, \lambda)$. Weak duality tells us that the dual objective value is upper bounded by the primal objective value.

$$\sup_{\lambda \in \mathbb{R}_+^{|\mathcal{C}|}} D(\lambda) \leq L(w_*) \tag{12}$$

As there is a feasible point and the convex constraints induce a polytope, Slater's condition is satisfied and strong duality holds. To analyze the utility of $\bar{w}$, we choose a setting of dual multipliers $\bar{\lambda} \in \mathbb{R}_+^{|C|}$ such that the duality gap $\gamma(w, \lambda) = L(w) - D(\lambda)$ is bounded (with high probability over the random choice of stochastic subgradients). Exhibiting such a setting of $\bar{\lambda}$ demonstrates the near optimality of $\bar{w}$.

**Lemma 7.** *Let $\tau, \delta > 0$ and $\mathcal{F} = [-B, B]^n$. After running Algorithm 1 for $T > \frac{30^2 M^2 B^2 n \log(n/\delta)}{\tau^2}$ iterations, then with probability at least $1 - 8\delta$ (over the stochastic subgradients)*

$$L(\bar{w}) \leq L(w_*) + \frac{3G}{5M}\tau.$$

We give the full proof of Lemma 7 in Appendix A.3.

## A.1 Answering residual queries

Next, we describe how to answer residual queries $R_S(w)$ efficiently, in terms of time and samples.

**Lemma 8.** *For $\tau, \gamma > 0$, for a $\gamma$-large collection of comparisons $\mathcal{C} \subseteq 2^{\mathcal{X} \times \mathcal{X}}$, with probability $1 - \delta$, given access to $n$ metric samples, every residual query $R_S(w)$ can be answered correctly with tolerance $\tau$ provided*

$$n \geq \tilde{\Omega}\left(\frac{\log(|C|/\delta)}{\gamma \cdot \tau^2}\right).$$

*Each residual query $R_S(w)$ can be answered after $\tilde{O}\left(\frac{\log(T \cdot |\mathcal{C}|/\delta)}{\gamma \cdot \tau^2}\right)$ evaluations of the current hypothesis.*

*Proof.* Recall the definition of $R_S(w)$.

$$R_S(w) = \mathop{\mathbf{E}}_{(x,x') \sim S}\left[|f_w(x) - f_w(x')|\right] - \mathop{\mathbf{E}}_{(x,x') \sim S}\left[d(x, x')\right]$$

Proposition 9 shows that $\mathbf{E}_S[d(x, x')]$ can be estimated for all $S \in \mathcal{C}$ from a small number of metric samples. The proof follows a standard Chernoff plus union bound argument. For completeness, we give a full proof next. Thus, Lemma 8 follows by showing that at each iteration $\mathbf{E}_S[|f_w(x) - f_w(x')|]$ can be estimated from a small number of evaluations of the current hypothesis $f_w$.

We can estimate the expected value of the deviation on $f$ over $S \in \mathcal{C}$ with a small set of unlabeled samples from $\mathcal{X} \times \mathcal{X}$; we will evaluate the hypothesis $f$ for each of these samples. Using an identical argument as in the case of the expected metric value, we can prove the following bound on how many comparisons we need to make, which shows the lemma.

**Proposition 9.** *Suppose $\mathcal{C}$ is $\gamma$-large. Then with probability at least $1 - \delta$, for all $S \in \mathcal{C}$, the empirical estimate for $\mathbf{E}_S[|f(x) - f(x')|]$ of $n$ samples $(x, x') \sim \mathcal{M}$ deviates from the true expected value by at most $\tau$ provided*

$$n \geq \tilde{\Omega}\left(\frac{B^2 \log(|C|/\delta)}{\gamma \cdot \tau^2}\right).$$

$\square$

Here, we show that a small number of samples from the metric suffices to estimate the expected metric distance over all $S \in \mathcal{C}$. Suppose $\mathcal{C}$ is $\gamma$-large. Then with probability at least $1 - \delta$, for all $S \in \mathcal{C}$, the empirical estimates for $\mathbf{E}_S[d(x_i, x_j)]$ of $n$ metric samples deviate from their true expected value by at most $\tau$ provided

$$n \geq \tilde{\Omega}\left(\frac{\log(|\mathcal{C}|/\delta)}{\gamma \cdot \tau^2}\right).$$

*Proof.* Let $(x_i, x_j, \Delta_{ij})$ represent a random metric sample. Suppose for each $S \in \mathcal{C}$, we obtain $m$ such samples where $(x_i, x_j) \sim S$, and let $d(S) = \sum_{ij \in X_S} \Delta_{ij}$ be the empirical average over the sample. Then, by Hoeffding's inequality, we know

$$\Pr\left[\left|d(S) - \mathop{\mathbf{E}}_{(x_i,x_j) \sim S}[d(x_i, x_j)]\right| > \tau\right] \leq 2e^{-2s\tau^2}.$$

If $m \geq \Omega\left(\frac{\log(|\mathcal{C}|/\delta)}{\tau^2}\right)$, then the probability that the estimate $d(S)$ is not within $\tau$ of its true value is less than $\frac{\delta}{|\mathcal{C}|}$. Union bounding over $\mathcal{C}$, the probability that every estimate has tolerance $\tau$ will be at least $1 - \delta$.

Because $\mathcal{C}$ is $\gamma$-large, for every $S \in \mathcal{C}$, the probability a random metric sample $(x_i, x_j) \sim \mathcal{M}$ is in $S$ is at least $\gamma$. If we take $\frac{\log(m)}{\gamma}$ samples, then with probability at least $1 - 1/m$, one of the samples will be in $S$. Thus, to guarantee $d(S)$ has tolerance $\tau$ for all $S \in \mathcal{C}$ with probability $1 - \delta$,

$$s = \frac{m \log(m)}{\gamma} = \tilde{\Omega}\left(\frac{\log(|\mathcal{C}|/\delta)}{\gamma \cdot \tau^2}\right)$$

samples suffice. □

## A.2 Answering subgradient queries

Next, we argue that the subgradient oracles can be implemented efficiently without accessing any metric samples. First, suppose we want to take a step according to $R_S(w)$; while $R_S(w)$ is not differentiable, we can compute a legal subgradient defined by partial subderivatives given as follows.

$$\frac{\partial R_S(w)}{\partial w_l} = \mathop{\mathbf{E}}_{(x,x')\sim S}[\mathbf{sgn}(\langle w, x - x' \rangle) \cdot (x_l - x'_l)] \tag{13}$$

The subgradient does not depend on $d$, so no samples from the metric are necessary. Further, Algorithm 1 only assumes access to stochastic subgradient oracle with bounded entries. If we sample a single $(x_i, x_j) \sim \mathcal{M}$, then $\mathbf{sgn}(\langle w, x_i - x_j \rangle) \cdot (x_{il} - x_{jl})$ will be an unbiased estimate of a subgradient of $R_S(w)$; we claim, the entries will also be bounded. In particular, assuming $\|x_i\|_1 \leq 1$ implies each partial is bounded by 2, so that we can take $M^2 = 4n$.

## A.3 Utility analysis of Algorithm 1

In this appendix, we give a full proof of Lemma 7. We defer the proof of certain technical lemmas to Appendix A.4 for the sake of presentation.

**Proof of Lemma 7** Let $\tau, \delta > 0$ and $\mathcal{F} = \{w \in \mathbb{R}^n : \|w\|_\infty \leq B\}$. After running Algorithm 1 for $T > \frac{30^2 M^2 B^2 n \log(n/\delta)}{\tau^2}$ iterations, then

$$L(\bar{w}) \leq L(w_*) + \frac{3G}{5M}\tau$$

with probability at least $1 - 8\delta$ over the randomness of the algorithm.

*Proof.* As before, we refer to $K_f \subseteq [T]$ as the set of feasible iterations, where we step according to the objective, and $[T] \setminus K_f$ as the set of infeasible iterations, where we step according to the violated constraints. Recall, we denote the set of subgradients of a function $L$ (or $R$) at $w$ by $\partial L(w)$ and denote by $\nabla L(w)$ a stochastic subgradient, where $\mathbf{E}[\nabla L(w)|w] \in \partial L(w)$.

When we do not step according to the objective, we step according to the subgradient of some violated comparison constraint. In fact, we show that stepping according to any convex combination of such subgradients suffices to guarantee progress in the duality gap. In the case wher $t \notin K_f$, we assume that we can find some convex combination $\sum_{S \in \mathcal{C}} \alpha_{k,S} \hat{R}_S(w_k) > 4\tau/5$ where for all $S \in \mathcal{C}, \alpha_{k,S} \in \Delta_{|\mathcal{C}|-1}$. We show that if we step according to the corresponding combination of the subgradients of $R_S(w_k)$, we can bound the duality gap. Specifically, for $k \notin K_f$, let the algorithm's step be given by

$$\sum_{S \in \mathcal{C}} \alpha_{k,S} \nabla R_S(w_k)$$

where for each $S \in \mathcal{C}$, we have $\mathbf{E}\left[\nabla R_S(w_k)|w_k\right] \in \partial R_S(w_k)$. Let $\eta_L = \frac{\tau}{GM}$ and $\eta_R = \frac{\tau}{M^2}$ denote the step size for the objective and residual steps, respectively. Then, consider the following choice of dual multipliers for each $S \in \mathcal{C}$.

$$\bar{\lambda}_S = \frac{\eta_R}{\eta_L |K_f|} \sum_{k \notin K_f} \alpha_{k,S} \tag{14}$$

Expanding the definition of $\bar{w}$ and applying convexity, we can bound the duality gap as follows

$$\gamma(\bar{w}, \bar{\lambda}) = L(\bar{w}) - D(\bar{\lambda}) \tag{15}$$

$$\leq \frac{1}{|K_f|} \left( \sum_{k \in K_f} L(w_k) \right) - \inf_{w \in \mathcal{F}} \left\{ L(w) + \sum_{S \in \mathcal{C}} \bar{\lambda}_S R_S(w) \right\} \tag{16}$$

$$= \sup_{w \in \mathcal{F}} \left\{ \frac{1}{|K_f|} \left( \sum_{k \in K_f} L(w_k) \right) - L(w) - \sum_{S \in \mathcal{C}} \bar{\lambda}_S R_S(w) \right\} \tag{17}$$

$$= \sup_{w \in \mathcal{F}} \left\{ \frac{1}{\eta_L |K_f|} \left( \eta_L \sum_{k \in K_f} (L(w_k) - L(w)) - \eta_R \sum_{k \notin K_f} \sum_{S \in \mathcal{C}} \alpha_{k,S} R_S(w) \right) \right\} \tag{18}$$

where (16) follows from expanding $\bar{w}$ then applying convexity of $L$ and the definition of $d(\bar{\lambda})$ and (18) follows by our choice of $\bar{\lambda}_S$ for each $S \in \mathcal{C}$.

With the duality gap expanded into one sum over the feasible iterates and one sum over the infeasible iterates, we can analyze these iterates separately. The following lemmas show how to track the contribution of each term to the duality gap in terms of a potential function $u_k$ defined as

$$u_k(w) = \frac{1}{2} \|w - w_k\|^2.$$

For notational convenience, for each $k \in K_f$, let $e(w_k) = \mathbf{E}[\nabla L(w_k) | w_k] - \nabla L(w_k)$ be the noise in the subgradient computation.

**Lemma 10.** *For all $w \in \mathcal{F}$ and for all $k \in K_f$,*

$$\eta_L \cdot (L(w_k) - L(w)) \leq u_k(w) - u_{k+1}(w) + \frac{\tau^2}{2M^2} + \eta_L \langle e(w_k), w_k - w \rangle.$$

Again, for notational convenience, for each $k \in [T] \setminus K_f$, let $e(w_k) = \sum_{S \in \mathcal{C}} \alpha_{k,S} \left( \mathbf{E}[\nabla R_S(w_k) | w_k] - \nabla R_S(w_k) \right)$ be the noise in the subgradient computation.

**Lemma 11.** *For all $w \in \mathcal{F}$ and for all $k \in [T] \setminus K_f$,*

$$-\eta_R \sum_{S \in \mathcal{C}} \alpha_{k,S} R_S(w) \leq u_k(w) - u_{k+1}(w) - \frac{\tau^2}{10M^2} + \eta_R \langle e(w_k), w_k - w \rangle.$$

We defer the proofs of Lemmas 10 and 11 to Appendix A.4. Assuming Lemmas 10 and 11, we bound the duality gap as follows.

$$\sup_{w \in \mathcal{F}} \left\{ \frac{1}{\eta_L |K_f|} \left( \sum_{k=1}^{T} [u_{k-1}(w) - u_k(w)] + \eta_L \sum_{k \in K_f} \langle e(w_k), w_k - w \rangle \right. \right.$$

$$\left. \left. + \eta_R \sum_{k \notin K_f} \langle e(w_k), w_k - w \rangle + \frac{\tau^2}{2M^2} |K_f| - \frac{\tau^2}{10M^2} (T - |K_f|) \right) \right\} \tag{19}$$

$$\leq \frac{1}{\eta_L |K_f|} \underbrace{\left( \sup_{w \in \mathcal{F}} \left\{ u_0(w) + \eta_L \sum_{k \in K_f} \langle e(w_k), w_k - w \rangle + \eta_R \sum_{k \notin K_f} \langle e(w_k), w_k - w \rangle \right\} - \frac{\tau^2}{10M^2} T \right)}_{(*)}$$

$$+ \underbrace{\frac{G\tau}{2M} + \frac{G\tau}{10M}}_{(**)} \tag{20}$$

by rearranging. Noting that $(**)$ can be bounded by $\frac{3G}{5M}\tau$, it remains to bound $(*)$. We show that for a sufficiently large $T$, then $(*)$ cannot be positive.

Consider the terms in the supremum over $w \in \mathcal{F}$. Note that we can upper bound $\sup\{u_0(w)\} \leq 2B^2 n$. Additionally, we upper bound the error incurred due to the objective subgradient noise with the following lemma, which we prove in Appendix A.4.

**Lemma 12.** *With probability at least $1 - 4\delta$, the contribution of the noisy subgradient computation to the duality gap can be bounded as follows.*

$$\sup_{w\in\mathcal{F}} \left\{ \eta_L \sum_{k\in K_f} \langle e(w_k), w_k - w \rangle + \eta_R \sum_{k\notin K_f} \langle e(w_k), w_k - w \rangle \right\} \leq \frac{\tau B}{M} \sqrt{8Tn\log(n/\delta)} \quad (21)$$

Thus, we can bound $(*)$ as follows.

$$(*) \leq 2B^2 n + \frac{\tau B}{M}\sqrt{8Tn\log(n/\delta)} - \frac{\tau^2}{10M^2}T$$

Assuming the lemma and that $T > \frac{30^2 M^2 B^2 n \log(n/\delta)}{\tau^2}$, then, we can bound $(*)$ by splitting the negative term involving $T$ to balance both positive terms.

$$(*) \leq \left( 2B^2 n - \frac{\tau^2}{10M^2}\cdot\frac{20T}{30^2} \right) + \left( \frac{\tau B}{M}\sqrt{8n\log(n/\delta)}\cdot\sqrt{T} - \frac{\tau^2}{10M^2}\cdot\frac{(30^2 - 20)T}{30^2} \right) \quad (22)$$

$$\leq \left( 2B^2 n - \frac{\tau^2}{10M^2}\frac{20M^2 B^2 n\log(n/\delta)}{\tau^2} \right)$$
$$+ \left( \frac{\tau B}{M}\sqrt{8n\log(n/\delta)}\cdot\frac{30MB\sqrt{n\log(n/\delta)}}{\tau} - \frac{\tau^2}{10M^2}\cdot\frac{(30^2-20)M^2 B^2 n\log(n/\delta)}{\tau^2} \right) \quad (23)$$

$$\leq \left( 2B^2 n - 2B^2 n\log(n/\delta) \right) + \left( 85B^2 n\log(n/\delta) - 88B^2 n\log(n/\delta) \right) \quad (24)$$

Thus, the sum of $(*)$ and $(**)$ is at most $\frac{3G}{5M}\tau$. $\qquad\square$

## A.4 Deferred proofs from analysis of Algorithm 1

**Technical lemma** First, we show a technical lemma that will be useful in analyzing the iterates' contributions to the duality gap. Recall our potential function $u_k : \mathcal{F} \to \mathbb{R}$.

$$u_k(w) = \frac{1}{2}\|w_k - w\|_2^2 \quad (25)$$

We show that the update rule $w_{k+1} \leftarrow \pi_{\mathcal{F}}(w_k - \eta_k g_k)$ implies the following inequality in terms of $\eta_k, g_k, u_k(w)$, and $u_{k+1}(w)$.

**Lemma 13.** *Suppose $w_{k+1} = \pi_{\mathcal{F}}(w_k - \eta_k g_k)$. Then, for all $w \in \mathcal{F}$,*

$$\eta_k\langle g_k, w_k - w \rangle \leq u_k(w) - u_{k+1}(w) + \frac{\eta_k^2}{2}\|g_k\|_2^2. \quad (26)$$

*Proof.* Consider the differentiable, convex function $B_k : \mathcal{F} \to \mathbb{R}$.

$$B_k(w) = \eta_k\langle g_k, w - w_k \rangle + \frac{1}{2}\|w - w_k\|_2^2 \quad (27)$$

$$\langle \nabla B_k(w_{k+1}), w - w_{k+1} \rangle = \langle \eta_k g_k + w_{k+1} - w_k, w - w_{k+1} \rangle \quad (28)$$
$$= \langle \pi_{\mathcal{F}}(w_k - \eta_k g_k) - (w_k - \eta_k g_k), w - \pi_{\mathcal{F}}(w_k - \eta_k g_k) \rangle \quad (29)$$
$$\geq 0 \quad (30)$$

where (29) follows by substituting the definition of $w_{k+1}$ twice; and (30) follows from the fact that for any closed convex set $\mathcal{F}$ and $w_0 \notin \mathcal{F}$,

$$\langle \pi_{\mathcal{F}}(w_0) - w_0, w - \pi_{\mathcal{F}}(w_0) \rangle \geq 0.$$

Rearranging (28) implies the following inequality holds for all $w \in \mathcal{F}$.

$$\langle \eta_k g_k + w_{k+1} - w_k, w - w_{k+1} \rangle \geq 0 \tag{31}$$
$$\iff \eta_k \langle g_k, w_{k+1} - w \rangle \leq \langle w_{k+1} - w_k, w - w_{k+1} \rangle \tag{32}$$

We will use the following technical identity to prove the lemma.

**Proposition 14.** *For all $w \in \mathcal{F}$,*

$$\langle w_{k+1} - w_k, w - w_{k+1} \rangle = u_k(w) - u_{k+1}(w) - \frac{1}{2} \|w_{k+1} - w_k\|^2.$$

*Proof.*

$$
\begin{aligned}
u_k(w) - u_{k+1}(w) &= \|w_k - w\|^2 - \|w_{k+1} - w\|^2 \\
&= \|w_k\|^2 + \|w\|^2 - \|w_{k+1}\|^2 - \|w\|^2 + 2\langle w_{k+1} - w_k, w \rangle \\
&= \|w_k\|^2 - \|w_{k+1}\|^2 + 2\langle w_{k+1} - w_k, w \rangle \\
&= \|w_k\|^2 - \|w_{k+1}\|^2 - 2\langle w_k - w_{k+1}, w_{k+1} \rangle + 2\langle w_{k+1} - w_k, w - w_{k+1} \rangle \\
&= \|w_{k+1} - w_k\|^2 + 2\langle w_{k+1} - w_k, w - w_{k+1} \rangle
\end{aligned}
$$

$\square$

Finally, we can show the inequality stated in the lemma.

$$\eta_k \langle g_k, w_k - w \rangle = \eta_k \langle g_k, (w_{k+1} + \eta g_k) - w \rangle \tag{33}$$
$$\leq \langle w_{k+1} - w_k, w - w_{k+1} \rangle + \eta_k^2 \|g_k\|_2^2 \tag{34}$$
$$\leq u_k(w) - u_{k+1}(w) - \frac{1}{2} \|w_{k+1} - w_k\|_2^2 + \eta_k^2 \|g_k\|_2^2 \tag{35}$$
$$= u_k(w) - u_{k+1}(w) + \frac{\eta_k^2}{2} \|g_k\|_2^2 \tag{36}$$

where (34) follows from (32); (35) follows by using Proposition 14 to write the expression in terms of $u_k$'s; and (36) follows by the gradient step $w_{k+1} - w_k = \eta_k g_k$. $\square$

**Proof of Lemma 10**  Here, we bound the contribution to the duality gap of each of the feasible iterations $k \in K_f$ as follows.

$$\eta_L \cdot (L(w_k) - L(w)) \leq u_k(w) - u_{k+1}(w) + \frac{\tau^2}{2M^2} + \eta_L \langle e_L(w_k), w_k - w \rangle \tag{37}$$

*Proof.* Let $e_L(w_k) = g_L(w_k) - \nabla L(w_k)$ where $g_L(w_k) = \mathbf{E}[\nabla L(w_k)] \in \partial L(w_k)$.

$$\eta_L \cdot (L(w_k) - L(w)) \leq \eta \langle g_L(w_k), w_k - w \rangle \tag{38}$$
$$\leq \eta \langle \nabla L(w_k) + e_L(w_k), w_k - w \rangle \tag{39}$$
$$\leq u_k(w) - u_{k+1}(w) + \frac{\eta_L^2}{2} \|\nabla L(w_k)\|_2^2 + \eta_L \langle e_L(w_k), w_k - w \rangle \tag{40}$$
$$\leq u_k(w) - u_{k+1}(w) + \frac{\tau^2}{2M^2} + \eta_L \langle e_L(w_k), w_k - w \rangle \tag{41}$$

where (39) follows by substituting $g_L$; (40) follows by expanding the inner product and applying Lemma 13 to the first term; (41) follows by our choice of $\eta_L = \tau/GM$. $\square$

**Proof of Lemma 11**  Here, we bound the contribution to the duality gap of each of the infeasible iterates $k \in [T] \setminus K_f$. We assume $\hat{R}_{S_k}(w_k)$ has tolerance $\tau/5$. Then we show

$$-\eta_R \sum_{S \in \mathcal{C}} \alpha_{k,S} R_S(w) \leq u_k(w) - u_{k+1}(w) - \frac{\tau^2}{10M^2} + \eta_R \langle e_R(w_k), w_k - w \rangle. \quad (42)$$

*Proof.* Recall, we let $e(w_k) = \sum_{S \in \mathcal{C}} \alpha_{k,S} \left( \mathbf{E}[\nabla R_S(w_k) | w_k] - \nabla R_S(w_k) \right)$. For each $S \in \mathcal{C}$, for any $g_S(w_k) \in \partial R_S(w_k)$, we can rewrite $-R_S(w)$ as follows.

$$-R_S(w) = R_S(w_k) - R_S(w) - R_S(w_k)$$
$$\leq \langle g_S(w_k), w_k - w \rangle - R_S(w_k)$$

Multiplying by $\eta_R$ and taking the convex combination of $S \in \mathcal{C}$ according to $\alpha_k$, we apply Lemma 13 to obtain the following inequality.

$$-\eta_R \sum_{S \in \mathcal{C}} \alpha_{k,S} R_S(w) \leq \eta_R \left\langle \sum_{S \in \mathcal{C}} \alpha_{k,S} \nabla R_S(w_k) + e_R(w_k), w_k - w \right\rangle - \eta_R \sum_{S \in \mathcal{C}} \alpha_{k,S} R_S(w_k)$$

$$(43)$$

$$\leq u_k(w) - u_{k+1}(w) + \frac{\eta_R^2}{2} \left\| \sum_{S \in \mathcal{C}} \alpha_{k,S} \nabla R_S(w_k) \right\|_2^2$$
$$- \eta_R \sum_{S \in \mathcal{C}} \alpha_{k,S} R_S(w_k) + \eta_R \langle e_R(w_k), w_k - w \rangle$$

$$(44)$$

$$\leq u_k(w) - u_{k+1}(w) + \frac{\tau^2}{2M^2} - \frac{\tau}{M^2} \cdot \sum_{S \in \mathcal{C}} \alpha_{k,S} R_S(w_k) + \eta_R \langle e_R(w_k), w_k - w \rangle$$

$$(45)$$

$$\leq u_k(w) - u_{k+1}(w) - \frac{\tau^2}{10M^2} + \eta_R \langle e_R(w_k), w_k - w \rangle \quad (46)$$

where (43) follows by substituting $\nabla R_S(w_k)$ for each $g_S(w_k)$ and the definition of $e_R(w_k)$; (44) follows by expanding the inner product and applying Lemma 13; (45) follows by our choice of $\eta_k = \tau^2/M^2$; (46) follows by the fact that when we update according to a constraint, we know $\sum_{S \in \mathcal{C}} \alpha_{k,S} \hat{R}_S(w_k) \geq 4\tau/5$ with tolerance $\tau/5$, so $\sum_{S \in \mathcal{C}} \alpha_{k,S} R_S(w_k) \geq 3\tau/5$.  □

**Proof of Lemma 12**  Here, we show that with probability at least $1 - 4\delta$, the contribution of the noisy subgradient computation to the duality gap can be bounded as follows.

$$\sup_{w \in \mathcal{F}} \left\{ \eta_L \sum_{k \in K_f} \langle e(w_k), w_k - w \rangle + \eta_R \sum_{k \notin K_f} \langle e(w_k), w_k - w \rangle \right\} \leq \frac{\tau B}{M} \sqrt{8Tn \log(n/\delta)} \quad (47)$$

*Proof.* Let $\varepsilon = \eta_L \cdot g = \eta_R \cdot m = \frac{\tau}{M\sqrt{n}}$. Further, let $\eta_k = \eta_L$ for $k \in K_f$ and $\eta_R$ for $k \notin K_f$. Then, we can rewrite the expression to bound using $\eta_k$ and expand as follows.

$$\sup_{w \in \mathcal{F}} \sum_{k \in [T]} \langle \eta_k e(w_k), w_k - w \rangle = \sup_{w \in \mathcal{F}} \sum_{k \in [T]} \sum_{l=1}^{n} \eta_k e(w_k)_l \cdot (w_k - w)_l$$

$$= \sum_{l=1}^{n} (w_k)_l \sum_{k \in [T]} \eta_k e(w_k)_l + \sum_{l=1}^{n} \sup_{(w)_l} \left\{ (w)_l \cdot \sum_{k \in [T]} \eta_k e(w_k)_l \right\}$$

Consider the second summation, and consider the summation inside the supremum. Note that this summation is a sum of mean-zero random variables, so it is also mean-zero. Recall, we assume the estimate of the $k$th subgradient is independent of the prior subgradients, given $w_k$. Further, by the

assumed $\ell_\infty$ bound on the subgradients, each of these random variables is bounded in magnitude by $\varepsilon$. Using the bounded difference property, we apply Azuma's inequality separately for each $l \in [n]$.

$$\Pr\left[\left|\sum_{k\in[T]}\eta_k e(w_k)_l\right| > Z\right] \le 2 \cdot \exp\left(-\frac{Z^2}{2T\varepsilon^2}\right)$$

Taking this probability to be at most $2\delta/n$, we can upper bound $Z$ by $\varepsilon\sqrt{2T\log(n/\delta)} = \frac{\tau}{M}\sqrt{2Tn\log(n/\delta)}$. Then, noting that $|(w)_l| < B$ for any $w \in \mathcal{F}$, we can take a union bound to conclude with probability at least $1 - 2\delta$ the following inequalities hold.

$$\sum_{l=1}^{n}\sup_{(w)_l}\left\{(w)_l \cdot \sum_{k\in[T]}\eta_k e(w_k)_l\right\} \le Bn \cdot Z$$

$$= \frac{\tau B}{M}\sqrt{2Tn\log(n/\delta)}$$

Further, we note that the first summation concentrates at least as quickly as the second, so by union bounding again,

$$\sup_{w\in\mathcal{F}}\sum_{k\in[T]}\eta_k \langle e(w_k), w_k - w\rangle \le \frac{\tau B}{M}\sqrt{8Tn\log(n/\delta)}$$

with probability at least $1 - 4\delta$. □

## B    Hardness for metric multifair predictions

The lower bounds follow the same general construction. Suppose we have some boolean concept class $\mathcal{C} \subseteq \{-1,1\}^\mathcal{X}$ for some universe $\mathcal{X}$. We will construct a new universe $\mathcal{X}_{01} = \mathcal{X}_0 \cup \mathcal{X}_1$ where $\mathcal{X}_0 \subseteq \mathcal{X}$ and define a collection of "bipartite" comparisons over subsets of $\mathcal{X}_0 \times \mathcal{X}_1$. We will assume access to random samples $(x_0, y(x_0))$ for some function $y : \mathcal{X} \to \{-1,1\}$; we define corresponding metric values where $d(x_0, x_1)$ is some function of $y(x_0)$ for all $x_1 \in \mathcal{X}_1$. Finally, we need to label $\mathcal{X}_{01}$ such that such that to obtain good loss, the post-processing algorithm must learn something non-trivial about $y$ (which will differ across the two lower bounds we prove).

Consider $\mathcal{X}_{01} = \mathcal{X}_0 \cup \mathcal{X}_1$ with ideal labels given as $(x_0, 0)$ for $x_0 \in \mathcal{X}_0$ and $(x_1, 1)$ for $x_1 \in \mathcal{X}_1$, and let $L$ be the hinge loss. We encode the original boolean concept in the distance metric, where for $x_0 \in \mathcal{X}_0$,

$$d(x_0, x_1) = 1 - y(x_0)$$

for all $x_1 \in \mathcal{X}_1$.

Then, consider the collection of comparisons given by $\mathcal{H} = \{S_c : c \in \mathcal{C}\}$ where we take $S_c = \{(x_0, x_1) \in \mathcal{X}_0 \times \mathcal{X}_1 : c(x_0) = 1\}$. We take $|\mathcal{X}_1| = 2|\mathcal{X}_0|$, large enough that the average prediction for $x_1 \in \mathcal{X}_1$ is at least $1-\tau$ in the optimal utility set of multifair predictions. In particular, any average deviation by more than $\tau$ in $\mathcal{X}_1$ would result in larger loss than setting all of $f(x_0) = f(x_1) = 1 - \tau$ for $x_1 \in \mathcal{X}_1$ and all $x_0 \in \mathcal{X}_0$ where $y(x_0) = 1$.

**Outline of sample complexity lower bound.**    Here, we outline the proof of Theorem 4

**Theorem** (Restatement of Theorem 4). *Let $\gamma, \tau > 0$ be constants and suppose $\mathcal{A}$ is an algorithm that has random sample access to $d$ and outputs a $(\mathcal{C}, d, \tau)$-metric multifair set of predictions for $\gamma$-large $\mathcal{C}$. Then, $\mathcal{A}$ takes $\Omega(\log|\mathcal{C}|)$ random samples from $d$ or outputs a set of predictions with loss that approaches the loss achievable with no metric queries.*

*Proof.* The theorem follows from our construction. In particular, if we take the concept class $\mathcal{C}$ on $\mathcal{X}_0$ to be set of linear functions over $\mathbb{F}_2$, and take $y = c$ to be a uniformly random $c \in \mathcal{C}$, then we get a hard distribution over metrics. Specifically, if the we take the concepts to be $n$-dimensional, then without $n$ linearly-independent queries to the metric, we will not be able to learn the concept with non-trivial accuracy. In particular, any algorithm that guarantees metric multifairness must assume that $\mathbf{E}_S[d(x_0, x_1)] \approx 0$ more than one $S \in \mathcal{H}$, which results in suboptimality. Thus, the assumption that the metric multifair learner achieved near-optimal utility must be false. □

Further note, if we only take $n - k$ queries, the incurred loss approaches the trivial loss exponentially in $k$. Appealing to lower bounds on the number of random queries needed to span a basis, the theorem follows.

**Outline of hardness from pseudorandom functions**   We use the construction to demonstrate that under weak complexity assumptions, there are algorithmic barriers to generally efficient algorithms for metric multifairness. In particular, we will assume that $\mathcal{C}$ defines a pseudorandom function family. The existence of one-way functions implies the existence of pseudorandom functions [11], so the proposition follows.

**Proposition** (Formal statement of Proposition 5). *Assuming one-way functions exist, any algorithm for computing $(\mathcal{H}, \tau)$-optimal $(\mathcal{H}, d, \tau)$-metric multifair predictions for any $\mathcal{H}, d,$ and $\tau > 0$ requires time $(\log |\mathcal{H}|)^{\omega(1)}$.*

*Proof.* Suppose we can post-process predictions to achieve $(\mathcal{C}, \tau)$-optimal $(\mathcal{C}, d, \tau)$-metric multifair predictions for any $\mathcal{C}$ and $d$ and some small constant $\tau > 0$. Let $\mathcal{C}$ define a pseudorandom function family. We will show that we can distinguish between a function $y = c \leftarrow_R \mathcal{C}$ drawn uniformly at random from the function family and a truly random function $y : \mathcal{X} \to \{-1, 1\}$.

We're given some samples of the form $(x, y(x)) \sim \mathcal{D} \times \{-1, 1\}$. Returning to the proposed construction, in the case $y$ is a truly random function, then with high probability, $\mathbf{E}_S[d(x_0, x_1)] \geq 1 - o(1)$ for all $S \in \mathcal{H}$. Thus, any $\tau$-optimal set of predictions will achieve loss $O(\tau)$.

In the case, where $y = c$ for some $c \in \mathcal{C}$, note that labeling $x_0 \in \mathcal{X}_0$ according to $c(x_0)$ and all $x_1 \in \mathcal{X}_1$ as 1 is a feasible point that obtains loss

$$\mathbf{E}_{x \sim \mathcal{X}_0}[L(f(x), 0)]/3 + 2 \cdot \mathbf{E}_{x \sim \mathcal{X}_1}[L(f(x), 1)]/3 = \Pr_{x_0 \sim \mathcal{X}_0}[c(x_0) = 1]/3$$

This feasible quantity upper bounds the optimal loss. Then, we can express the expectation of difference in the predictions across the set defined by $c$ as follows.

$$\mathbf{E}_{(x_0, x_1) \sim S_c}\Big[|f(x_0) - f(x_1)| - (1 - c(x_0))\Big] = \mathbf{E}_{(x_0, x_1) \sim S_c}\Big[|f(x_0) - f(x_1)|\Big] - \mathbf{E}_{(x_0, x_1) \sim S_c}\Big[(1 - c(x_0))\Big]$$
$$\geq \mathbf{E}_{x_0 : c(x_0) = 1}[|f(x_0) - f(x_1)|] + 0.$$

We assume the set of predictions $f$ is $(\mathcal{H}, d, \tau)$-metric multifair. Thus, because $S_c \in \mathcal{H}$, we can upper bound this term by $\tau$. In total then, $\mathbf{E}_{x_0 : c(x_0) = 1}[|f(x_0) - f(x_1)|] \leq \tau$, and by the fact that $\mathbf{E}_{x_1 \sim \mathcal{X}_1}[f(x_1)] \geq 1 - \tau$, then $\mathbf{E}_{x_0 : c(x_0) = 1}[f(x_0)] \geq 1 - 2\tau$. Further, consider the following lower bound on the loss on $f$.

$$3 \cdot \mathbf{E}_{(x, y) \sim \mathcal{X} \times [-1, 1]}\big[ \max\{0, |f(x) - y|\} \big]$$
$$\geq \mathbf{E}_{x_0 \sim \mathcal{D}}[f(x_0)]$$
$$\geq \Pr_{x_0 \sim \mathcal{D}}[c(x_0) = 1] \cdot (1 - 2\tau)$$
$$\geq \Pr_{x_0 \sim \mathcal{X}_0}[c(x_0) = 1] - 2\tau$$

If $\mathcal{C}$ is a pseudorandom function family the $\Pr_{x_0 \sim \mathcal{X}_0}[c(x_0) = 1] \approx 1/2$ must be bounded away from 0. Thus, in the case where $y \in \mathcal{C}$, we have a non-trivial lower bound on the achievable loss under $(\mathcal{C}, d, \tau)$-metric multifairness. Thus, we can distinguish when $y \in \mathcal{C}$ and when $y$ is truly random.   $\square$

## Footnotes

[7]Such a $w^*$ exists, as $w = 0 \in \mathbb{R}^n$ always trivially satisfies all the fairness constraints.