[Reviews · NeurIPS 2018]

Reviewer 1



The paper builds on the influential "Fairness Through Awareness" paper of Dwork et al., which provided a framework for learning non-discriminatory classifiers by requiring the the classifier to treat similar individuals similarly. This similarity was defined by a (hypothetical) task-specific metric for determining the degree to which individuals are similar with respect to the classification task at hand. This paper extends the results to the more realistic scenario when the entire metric is not known to the learner, while relaxing the requirement that all pairs of similar individuals be treated similarly. The paper provides both learning algorithms and hardness results for this model. Fairness in machine learning is still a very young field, with many competing models and definitions, and more questions than answers. In this area, asking the right basic questions, usually in the form of fairness definitions, is still of crucial importance. The Dwork et al. paper is one of the original seminal papers of the area, but it had weaknesses, first and foremost the requirement of access to a task-specific similarity metric for individuals. This paper, by relaxing this assumption as well as the fairness constraints, extends the results of Dwork et al. to much more realistic and fine-grained scenarios. The new definitions in this paper are clever and thoughtful, and the reasoning behind them is explained clearly with examples and counterexamples. The theorems are nontrivial, stated precisely and explained clearly. Overall this is a high quality paper with significant novel contributions to the theory of fair learning. The main weakness of the paper is the lack of empirical results. It seems that no experiments were done to validate the model on real-world data. It would be helpful to explain this lack of empirical results. In particular, if it is because this more realistic model is still not quite realistic enough to be used in practical applications, I would prefer that the authors honestly acknowledge this fact and discuss what the roadblocks are and what is needed for this framework to become applicable - whether it's further algorithmic advances or the availability of specific kinds of data.

Reviewer 2



The authors propose a new notion of fairness called metric multifairness and show how to achieve it, i.e. "treat similar sub-populations of individuals similarly" The intuition and rationale behind the fairness through computationally-bounded awareness principle are well-explained and can be summarized as "while information-theoretically, we can’t hope to ensure fair treatment across all subpopulations, we can hope ensure fair treatment across efficiently-identifiable subpopulations." My main concern is that this application driven work does not provide realistic examples of how the proposed metric and approach work be used. An experimental assessment of the approach would be valuable in this regard as well. Minor: some typos, e.g. mteric -> metric -- I read the author response

Reviewer 3



The paper aims to extend the "fairness through awareness" framework by assuming that instead of having access to a task-specific distance metric on individuals, the decision-maker can make a small number of queries to an expert panel which could answer a small number of queries. The authors generalize the notion of metric fairness to "metric multifairness," which considers a collection of sets of pairs of individuals and requires that for each such set, the expected difference in scores given to the pairs in that set is not much more than the expected distance between pairs. They show that if this collection of sets is expressive, then this property implies that a large fraction of pairs of individuals satisfy metric fairness. As a result, the authors propose to consider a family defined by some bounded computation to make the sets considered sufficiently rich while also maintaining the ability to efficiently sample the distance metric. The authors show how to cast learning a linear classifier under metric multifairness constraints as a convex problem and provide convergence guarantees. They also give a postprocessing technique to satisfy metric multifairness for an arbitrary given classifier. They go on to give a reduction to agnostic learning, showing that if the concept and hypothesis classes can be learned by an agnostic learner, then this can be used to make more efficient use of the metric samples. Finally, they show that their bounds are tight up to constant factors, and that some learnability assumptions are necessary to make any claims about utility. The premise of this work is fairly interesting -- learning a metric using guidelines from an expert panel is a compelling idea. On the other hand, it's unclear that the guarantees that they achieve are very strong. It would behelpful to see some explicit examples of concept classes and the particular guarantees they would imply. In addition, it would be good to see experimentally how the guarantee from Proposition 1 plays out in practice: does it result in strong individual fairness for a constant fraction of the population, or does it afford a slightly weaker protection to almost all of the population? Overall, I think this work is reasonably good, but in order to have stronger feelings about it I'd need to see a more concrete instantiation of the class of comparisons and perhaps some (minor) experimental results demonstrating the effects of metric multifairness. In particular, I don't have a good sense for how many metric queries are needed in practice -- is it something like 100? 1000? While the theory behind this work is nice, to me, it isn't that interesting on its own, and would be made stronger by having some empirical demonstrations to back it up.

Reviewer 4



The paper studies the problem of fair classification in the framework of fairness through awareness" of Dwork et al which requires similar individuals to be labeled similarly by the classifier. Crucially, the work of Dwork et al requires the existence of a similarity metric between the pairs of individuals; an assumption that is one of the most challenging aspects" of the framework. This work relaxes this assumption by requiring similar identifiable subpopulations to be treated similarly by the classifier. The authors assume while the metric is not known, it can be queried for specific pairs of individuals by the learner. The authors show under what conditions this relaxed notion of fairness can be efficiently satisfied when the learner's goal is to maximize classification's utility subject to satisfying this new relaxed fairness constraint. Questions: (1) Does the post-processing approach in section 3.1 come with any formal theoretical guarantees? Woodworth et al 2017 have shown that post-processing to satisfy equality of odds can be highly sub-optimal in the worst case. Strengths: (1) While the paper is similar in spirit to the work of Hebert-Johnson et al and Kearns et al, I feel like relaxing the fairness through awareness framework is a rather interesting contribution. (2) The paper is rather well-written and the high level ideas are easy to follow. Weaknesses: (1) If I understand correctly, the authors assume a realizable setting for the generation of labeled examples i.e. the unlabeled examples are generated by a distribution D and then the unlabeled examples are labeled by a deterministic function. Can the authors clarify if this is the correct understanding? If so, given that the realizability assumption is rather restrictive/outdated, can the authors elaborate how/whether their results can be extended to an unrealizable setting? (2) The authors draw connections to agnostic learning problems to solve the search step of algorithm 1. Agnostic learning is generally hard computationally but regardless have been commonly used/solved in practices. While I strongly believe that theoretical contributions are indeed important, in a topic such as an algorithmic fairness, it would be rather crucial to see how the techniques proposed in this paper fare when dealing with real datasets even if heuristic techniques are used to solve the computationally hard subroutines. (3) The assumption that the hypothesis class is the class of linear functions is rather restrictive. As far as I can understand the constraint in the convex program might not be convex if a different hypothesis class was used. Typos and minor comments: (1) line (17): attributes (2) line (55): requiring (3) line (83): linear function "with" bounded weights (4) No "." needed in the middle of line 247 (5) typo in line 292/293 -- last sentence of the paragraph (6) the paper uses rather unconventional notation for the ML audience: for example in a typical ML paper delta is used for confidence parameter, epsilon is used for accuracy, n/d is used for dimension and m is used for sample complexity. While this is a matter of choice, the paper would be much easier to read for a NIPS/ICML audience if the standard notation were followed. ------------------------------------------------------------- I read the other reviews and the author response.